# SELF-SUPERVISE, REFINE, REPEAT: IMPROVING UN-SUPERVISED ANOMALY DETECTION

## ABSTRACT

Anomaly detection (AD) – separating anomalies from normal data – has many applications across domains, from manufacturing to healthcare. While most previous works have been shown to be effective for cases with fully or partially labeled data, that setting is in practice less common due to labeling being particularly tedious for this task. In this paper, we focus on fully *unsupervised* AD, in which the entire training dataset, containing both normal and anomalous samples, is unlabeled. To tackle this problem effectively, we propose to improve the robustness of one-class classification trained on self-supervised representations using a data refinement process. Our proposed data refinement approach is based on an ensemble of one-class classifiers (OCCs), each of which is trained on a disjoint subset of training data. Representations learned by self-supervised learning on the refined data are iteratively updated as the refinement improves. We demonstrate our method on various unsupervised AD tasks with image and tabular data. With a 10% anomaly ratio on CIFAR-10 image data / 2.5% anomaly ratio on Thyroid tabular data, the proposed method outperforms the state-of-the-art one-class classification method by 6.3 AUC and 12.5 average precision / 22.9 F1-score.

## 1 INTRODUCTION

Anomaly detection (AD), the task of distinguishing anomalies from normal data, plays a crucial role in many real-world applications such as detecting faulty products using visual sensors in manufacturing, fraudulent behaviors at credit card transactions, or adversarial outcomes at intensive care units.

AD has been considered under various settings based on the availability of negative (normal) and positive (anomalous) data and their labels at training, as overviewed in Sec. 2. Each application scenario is dominated by different challenges. When entire positive and negative data are available along with their labels (Fig. 1a), the problem can be treated as supervised classification and the dominant challenge becomes the imbalance in label distributions [4, 16, 20, 27, 32, 35]. When only negative labeled data are available (Fig. 1b), the problem is 'one-class classification' [23, 26, 33, 45, 47, 51, 53]. Various works have also extended approaches designed for these to settings with additional unlabeled data (Fig. 1c,d,e) [10, 18, 24, 46, 58] in a semi-supervised setting. While there exist many prior works in these settings, *they all depend on some labeled data*, which is not desirable in all application scenarios.

Unsupervised AD, on the other hand, poses unique challenges in the absence of any labeled data information, and a straightforward adaption of methods developed with the assumption of labeled data would be suboptimal. For example, some recent studies [7, 60] have applied one-class classifiers (OCCs) that are known to yield impressive performance when trained on negative samples [7, 23, 26, 33, 51] to unsupervised AD, but their performance for unsupervised AD has been quite sub-optimal. Fig. 2 illustrates this, showing the unsupervised AD performance of state-of-the-art Deep OCCs [51] with different anomaly ratios in unlabeled training data – the average precision significantly drops even when a small portion (2%) of training data is contaminated with anomalies.

Our framework SRR (**S**elf-supervise, **R**efine, **R**epeat), overviewed in Fig. 3, brings a novel approach to unsupervised AD with the principles of self-supervised learning without labels and iterative data refinement based on the agreement of OCC outputs. We propose to improve the state-of-the-art performance of OCCs, e.g. [33, 51], by refining the unlabeled training data so as to address the fundamental challenges elaborated above. SRR iteratively trains deep representations using refined

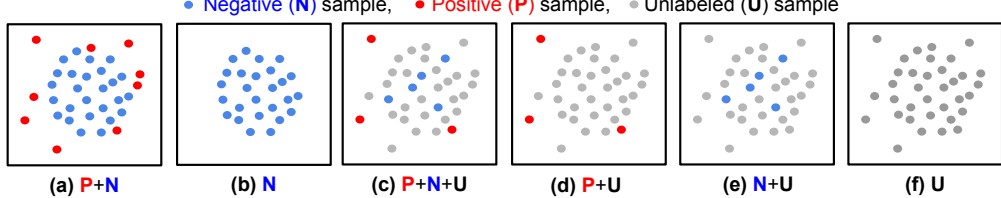

Figure 1: AD problem settings. Blue and red dots are for **labeled** negative (normal) and positive (anomalous) samples, respectively. Grey dots denote **unlabeled** samples. While previous works mostly focus on supervised (a, b) or semi-supervised (c, d, e) settings, we tackle an AD problem using only unlabeled data (f) that may contain both negative and positive samples.

data while improving the refinement of unlabeled data by excluding potentially-positive (anomalous) samples. For the data refinement process, we employ an ensemble of OCCs, each of which is trained on a disjoint subset of unlabeled training data. The samples are declared as normal if there is a consensus between all the OCCs. The refined training data are used to train the final OCC to generate the anomaly scores in the unsupervised setting. Most prior unsupervised AD works [23, 26, 33, 51] assume that the data contains entirely negative samples, which makes them not truly unsupervised as they require having humans to do the data filtering. Similar to ours, some prior unsupervised AD works [7, 45, 60] considered evaluating on an unsupervised setting where there exist a small percentage of anomalous samples in the training data, i.e. operating in 'truly' unsupervised setting without having the need for humans to do any filtering in the training data. However, these methods often suffered from significant performance degradation as the ratio of anomalous sample ratio has increased (see Sec. 4.2). We would like to highlight that our method distinguishes from the literature by bringing a data-centric approach (refining the unlabeled data) to unsupervised anomaly detection beyond the model-centric approaches (improving the model itself). Our framework SRR aims to provide robustness in performance as the anomalous sample ratio increases, as shown in Fig. 2.

We conduct extensive experiments across various datasets from different domains, including semantic AD (CIFAR-10 [29], Dog-vs-Cat [19]), real-world manufacturing visual AD use case (MVTec [8]), and tabular AD benchmarks. We consider methods with both shallow [36, 47] and deep [7, 33, 51] models. We evaluate models at different anomaly ratios of unlabeled training data and show that SRR significantly boosts performance. For example, in Fig. 2, SRR improves more than 15.0 average precision (AP) with a 10% anomaly ratio compared to a state-of-the-art one-class contrastive representation model [51] on CIFAR-10. Similarly, on MVTec SRR retains a strong performance, dropping less than 1.0 AUC with 10% anomaly ratio, while the best existing OCC [33] drops more than 6.0 AUC. We further investigate the efficacy of our design choices, such as the number of ensemble classifiers, thresholds, and refinement of deep representations via ablation studies.

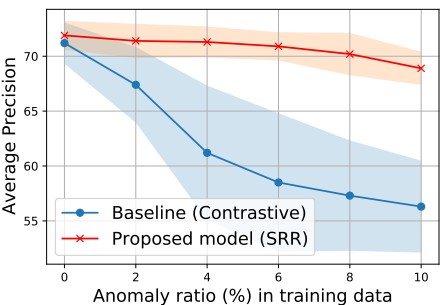

Figure 2: Performance of our proposed model and a baseline OCC using contrastive representation [51] on CIFAR-10 with different anomaly ratios in the training data.

## 2 RELATED WORK

There are various existing works under the different settings described in Fig. 1:

**The positive + negative setting** is often considered as a supervised binary classification problem. The challenge arises due to the imbalance in label distributions as positive (anomalous) samples are rare. As summarized in [12], to address this, over-/under-sampling [16, 20], weighted optimization [4, 27], synthesizing data of minority classes [32, 35], and hybrid methods [2, 22] have been studied.

**The negative setting** is often converted to a one-class classification problem, with the goal of finding a decision boundary that includes as many one-class samples as possible. Shallow models for this setting include one-class support vector machines [47] (OC-SVM), support vector data

description [53] (SVDD), kernel density estimation (KDE) [31], and Gaussian density estimation (GDE) [44]. There are also auto-encoder based models [59] that treat the reconstruction error as the anomaly score. Deep learning based OCCs have been developed, such as Deep OCC [45], geometric transformation [23], or outlier exposure [26]. Noting the degeneracy or inconsistency of learning objectives of existing end-to-end trainable Deep OCCs, [51] proposed a deep representation OCC, a two-stage framework that learns self-supervised representations [17, 28] followed by shallow OCCs. That work was extended for texture anomaly localization with CutPaste [33]. The robustness to very low anomaly ratios of these methods under the unsupervised setting was explored in [7, 60].

**The semi-supervised setting** is defined as utilizing a small set of labeled samples and large set of unlabeled samples to distinguish anomalies from normal data. Depending on which labeled samples are given, this setting can be split into three sub-categories. When only some positive/negative labeled samples are provided, we denote that as a PU/NU setting. Most previous works in semi-supervised AD settings focus on the NU setting where only some of the normal labeled samples are given [1, 41, 52]. The PNU setting is a more general semi-supervised setting where subsets of both positive and negative labeled samples are given. Deep SAD [46] and SU-IDS [39] are included in this category. We show the significant outperformance of our proposed SRR framework compared to Deep SAD, for multiple benchmark datasets when not using any labeled data (see Sec. 4.2).

**The unlabeled setting** has received relatively less attention despite its significance in automating machine learning. The popular methods for this setting include isolation forest [36] and local outlier factor [13]. However, they are difficult to scale, and less compatible with recent advances in representation learning. While OCCs, such as OC-SVM, SVDD, or their deep counterparts, apply to unlabeled settings by assuming the data is all negative, and the robustness of those methods has also been demonstrated in part [7, 60], in practice we observe a significant performance drop with a high anomaly ratio, shown in Fig. 2. In contrast, our proposed framework is able to maintain high performance across anomaly ratios.

**Data refinement** has been applied to AD in some prior works. [5, 42] generate pseudo-labels using binary classification and OC-SVM for data refinement to boost the consequent AD performances in unsupervised settings. [6, 30, 40, 55, 59] used the reconstruction errors of the auto-encoder as an indicator for removing possible anomalies. [21] and [38] used data refinement for AD in supervised and semi-supervised settings. Additional discussions can be found in Appendix A.9.

**Self-training** [37, 48] is an iterative training mechanism using predicted pseudo labels as targets for model training. It has regained popularity recently with its successful results in semi-supervised image classification [9, 50, 57]. To improve the quality of pseudo labels, employment of an ensemble of classifiers has also been studied. [14] trains an ensemble of classifiers with different classification methods to make a consensus for noisy label verification. Co-training [11] trains multiple classifiers, each of which is trained on the distinct views, to supervise other classifiers. Co-teaching [25] and DivideMix [34] share a similar idea in that they both train multiple deep networks on separate data batches to learn different decision boundaries, thus becoming useful for noisy label verification. While sharing a similarity, the proposed framework has clear differences from the previous works. SRR performs *iterative training* with data refinement (with robust *ensemble* methods) and self-supervised learning for *unsupervised* learning of an anomaly detector.

## 3 PROPOSED FRAMEWORK

**S**elf-supervise, **R**efine, and **R**epeat (SRR) is an iterative training framework, where we refine the data (Sec. 3.1) and update the representation with the refined data (Sec. 3.2), followed by OCCs on refined representations. Fig. 3 overviews the framework and Algorithm 1 provides the pseudo code.

**Notation.** We denote the training data as $\mathcal{D} = \{\mathbf{x}_i\}_{i=1}^N$ where $\mathbf{x}_i \in \mathcal{X}$ and $N$ is the number of training samples. $y_i \in \{0, 1\}$ is the corresponding label to $\mathbf{x}_i$, where 0 denotes normal (negative) and 1 denotes anomaly (positive). Note that labels are not provided in the unsupervised setting.

Let us denote a feature extractor as $g : \mathcal{X} \to \mathcal{Z}$. $g$ may include any data preprocessing functions, an identity function (if raw data is directly used for one-class classification), and learned or learnable representation extractors such as deep neural networks. Let us define an OCC as $f : \mathcal{Z} \to [-\infty, \infty]$ that outputs anomaly scores given the input features $g(\mathbf{x})$. The higher the score $f(g(\mathbf{x}))$, the more anomalous the sample $\mathbf{x}$ is. The binary anomaly prediction is made after thresholding: $\mathbb{1}\big(f(g(\mathbf{x})) \geq \eta\big)$.

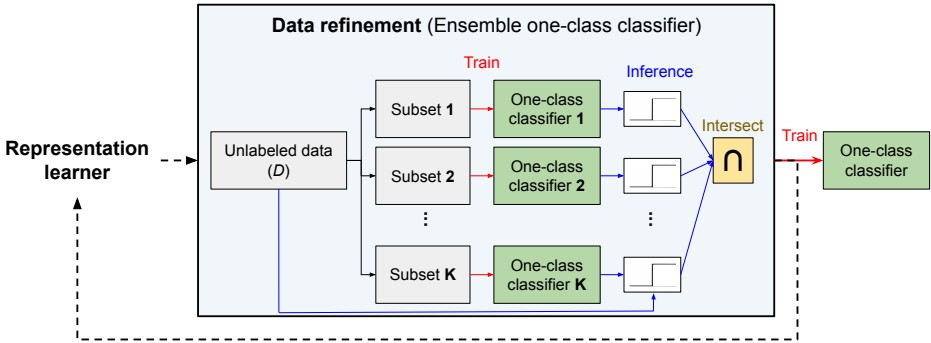

Figure 3: Block diagram of SRR composed of representation learner (Sec. 3.2), data refinement (Sec. 3.1), and final OCC blocks. The representation learner updates the deep models using refined data from the data refinement block. Data refinement is done by an ensemble of OCCs, each of which is trained on $K$ disjoint subsets of unlabeled training data. Samples predicted as normal by all classifiers are retained in the refined data, and are used to update the representation learner and final OCC. The process is repeated iteratively until convergence. Convergence graphs can be found in the Appendix A.8.

## 3.1 DATA REFINEMENT

A naive way to generate pseudo labels of unlabeled data is to construct an OCC on raw data or learned representations as in [51] and threshold the anomaly score to obtain a binary label for normal vs. anomalous. As we update the model with refined data that excludes samples that are predicted to be anomalous, it is important to generate pseudo labels of training data as accurately as possible.

To this end, instead of training a single classifier (unlike most previous works on data refinement for AD [5, 42, 59]), we train an ensemble of $K$ OCCs and aggregate their predictions to generate pseudo labels. We illustrate the data refinement block in Fig. 3 and as REFINEDATA in Algorithm 1. Specifically, we randomly divide the unlabeled training data $\mathcal{D}$ into $K$ disjoint subsets $\mathcal{D}_1, ..., \mathcal{D}_K$, and train $K$ different OCCs $(f_1, ..., f_K)$ on corresponding subsets $(\mathcal{D}_1, ..., \mathcal{D}_K)$. Then, we estimate a binary pseudo-label of the data $\mathbf{x}_i \in \mathcal{D}$ as follows:

$$\hat{y}_i = 1 - \prod_{k=1}^{K} \left[ 1 - \mathbb{1}\big(f_k(g(\mathbf{x}_i)) \geq \eta_k\big) \right] \tag{1}$$

$$\eta_k = \max \eta \;\; \text{s.t.} \;\; \frac{1}{N} \sum_{i=1}^{N} \mathbb{1}\big(f_k(g(\mathbf{x}_i)) \geq \eta\big) \geq \gamma \tag{2}$$

where $\mathbb{1}(\cdot)$ is the indicator function that outputs 1/0 if the input is True/False. $f_k(g(\mathbf{x}_i))$ represents an anomaly score of $\mathbf{x}_i$ for an OCC $f_k$. $\eta_k$ in equation 2 is a threshold determined as a $\gamma$ percentile of the anomaly score distribution $\{f_k(g(\mathbf{x}_i))\}_{i=1}^{N}$.

To interpret equation 1, $\mathbf{x}_i$ is predicted as normal, i.e. $\hat{y}_i = 0$, if all $K$ OCCs predict it as normal. While this may be too strict and potentially reject many true normal samples in the training set, we find that empirically, it is critical to be able to exclude true anomalous samples from the training set. The effectiveness of the employment of an ensemble of classifiers is empirically shown in Sec. 4.3.

## 3.2 REPRESENTATION UPDATE

SRR follows the idea of deep representation OCCs [51], where in the first stage a deep neural network is trained with self-supervised learning (such as rotation prediction [23], contrastive [51], or CutPaste [33]) to obtain meaningful representations of the data, and in the second stage OCCs are trained on these learned representations. Such a two-stage framework is shown to be beneficial as it prevents the 'hypersphere collapse' of the deep OCCs by the favorable inductive bias it brings with the architectural constraints [45].

Here, we propose to conduct self-supervised representation learning jointly with data refinement. More precisely, we train a feature extractor $g$ using $\hat{\mathcal{D}} = \{\mathbf{x}_i \,|\, \hat{y}_i = 0\}$, a subset of unlabeled data

---

**Algorithm 1** SRR: Self-supervise, Refine, Repeat.

---

**Input**: Training data $\mathcal{D} = \{\mathbf{x}_i\}_{i=1}^N$, Ensemble count ($K$), threshold ($\gamma$)
**Output**: Refined data ($\hat{\mathcal{D}}$), trained OCC ($f$), feature extractor ($g$)

1: **function** REFINEDATA($\mathcal{D}, g, K, \gamma$)
2:     Train OCC models $\{f_k\}_{k=1}^K$ on $\{\mathcal{D}_k\}_{k=1}^K$, $K$ disjoint subsets of the training data $\mathcal{D}$.
3:     Compute thresholds $\eta_k$'s for $\gamma$ percentile of anomaly distributions (equation 2).
4:     Predict binary labels $\hat{y}_i$ (equation 1).
5:     Return $\hat{\mathcal{D}} = \{\mathbf{x}_i : \hat{y}_i = 0, \mathbf{x}_i \in \mathcal{D}\}$.
6: **end function**
7: **function** SRR($\mathcal{D}, K, \gamma$)
8:     Initialize the feature extractor $g$.
9:     **while** $g$ not converged **do**
10:         $\hat{\mathcal{D}}$ = REFINEDATA($\mathcal{D}, g, K, \gamma$).
11:         Update $g$ using $\hat{\mathcal{D}}$ with self-supervised learning objectives.
12:     **end while**
13:     $\hat{\mathcal{D}}$ = REFINEDATA($\mathcal{D}, g, K, \gamma$).
14:     Train an OCC model ($f$) on refined data ($\hat{\mathcal{D}}$).
15: **end function**

---

$\mathcal{D}$ that only includes samples whose predicted labels with an ensemble OCC from Sec. 3.1 are negative. We also update $\hat{\mathcal{D}}$ as we proceed with representation learning. The proposed method is illustrated in Algorithm 1 as SRR. In contrast to previous works [33, 51] that use the entire training data for learning self-supervised representation, we find it necessary to refine the training data even for learning deep representations. Without representation refinement, the performance improvements of SRR are limited, as shown in Sec. 4.3.2. Last, for test-time prediction, we train an OCC on refined data $\hat{\mathcal{D}}$ using updated representations by $g$ as in line 13-14 in Algorithm 1.

### 3.3 UNSUPERVISED MODEL SELECTION

As SRR is designed for unsupervised AD, labeled validation data for hyperparameter tuning is typically not available and *the framework should enable robust model selection without any reliance on labeled data*. Here, we provide insights on how to select important hyperparameters, and later in Sec. 4.3.1 perform sensitivity analyses for these hyperparameters.

Data refinement of SRR introduces two hyperparameters: the number of OCCs ($K$) and the percentile threshold ($\gamma$). There is a trade-off between the number of classifiers for the ensemble and the size of disjoint subsets for training each classifier. With large $K$, we aggregate prediction from many classifiers, each of which may contain randomness from training. This comes at a cost of reduced performance per classifier as we use smaller subsets to train them. In practice, we find $K = 5$ works well across different datasets and anomaly ratios. $\gamma$ controls the purity and coverage of refined data. If $\gamma$ is large, and thus classifiers reject too many samples, the refined data could be more pure and contain mostly the normal samples; however, the coverage of the normal samples would be limited. On the other hand, with a small $\gamma$, the refined data may still contain many anomalies and the performance improvement with SRR would be limited. We empirically observe that SRR is robust to the selection of $\gamma$ when it is chosen from a reasonable range. In our empirical experiments, we find $\sim 1 - 2\times$ of the true anomaly ratio to be a reasonable choice. In other words, it is safer to use $\gamma$ higher than the expected true anomaly ratio. In some cases, the true anomaly ratio may not be available at all; for such scenarios, we propose Otsu's method [49] to estimate the anomaly ratio of the training data for determining the threshold $\gamma$ (experimental results are in the Appendix A.5).

## 4 EXPERIMENTS

We evaluate the efficacy of our proposed framework for unsupervised AD tasks on tabular (Sec. 4.1) and image (Sec. 4.2) data types. We experiment varying ratios of anomaly samples in unlabeled training data and with different combinations of representation learning and OCCs. In Sec. 4.3, we

provide performance analyses to better explain major constituents of the performance, as well as sensitivity to hyperparameter values.

**Implementation details:** To reduce the computational complexity of the data refinement block, we utilize a simple OCC such as GDE in the data refinement block. In a two-stage model, we only update the data refinement block at 1st, 2nd, 5th, 10th, 20th, 50th, 100th, 500th epochs instead of every epoch. After 500 epochs, we update the data refinement block per each 500th epoch. Each run of experiments requires a single V100 GPU. Additional discussions can be found in Appendix A.10.

## 4.1 EXPERIMENTS ON TABULAR DATA

**Datasets.** Following [7, 60], we test the efficacy of SRR on a variety of tabular datasets, including KDDCup, Thyroid, or Arrhythmia from the UCI repository [3]. We also use KDDCup-Rev, where the labels of KDDCup are reversed so that an attack represents anomaly [60]. To construct data splits, we use 50% of normal samples for training. In addition, we hold out some anomaly samples (amounting to 10% of the normal samples) from the data. This allows us to simulate unsupervised settings with an anomaly ratio of up to 10% of entire training set. The rest of the data is used for testing.[1] We conduct experiments using 5 random splits and 5 random seeds, and report the average and standard deviation of 25 F1-scores (with scale 0-100) for the performance metric.

**Models.** We mainly compare with GOAD [7] (the state-of-the-art AD model in the tabular domain) and implement SRR on top of it. GOAD utilizes random transformation classification as the pretext task of self-supervised learning, and the normality score is determined by whether transformations are accurately included in the transformed space of the normal samples. We re-implement GOAD [7] with a few modifications. First, instead of using embeddings to compute the loss, we use a parametric classifier, similarly to augmentation prediction [51]. Second, we follow the two-stage framework [51] to construct deep OCCs. For the clean training data setting, our implementation achieves 98.0 for KDD, 95.0 for KDD-Rev, 75.1 for Thyroid, and 54.8 for Arrhythmia F1-scores, which are comparable to those reported in [7]. Please see Appendix A.3 for formulation and implementation details.

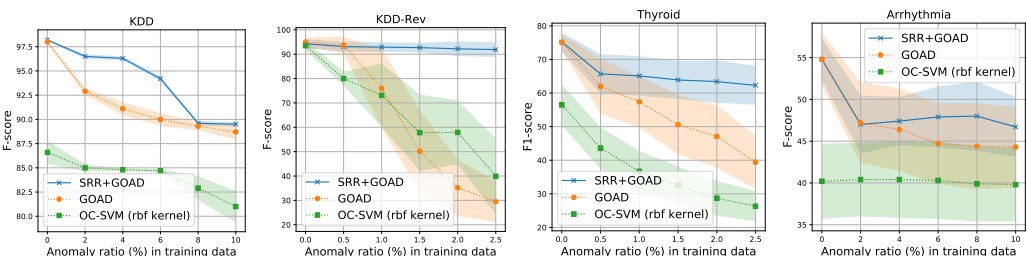

Figure 4: Unsupervised AD performance (F1-score) using OC-SVM (with rbf kernel), GOAD [7], and GOAD with the proposed method SRR on various tabular datasets. Shaded areas represent the standard deviation.

**Results.** We show results of GOAD (the baseline) and GOAD with SRR in Fig. 4. The ranges of the noise ratio are set to 0% for the anomaly ratios in the original dataset (if anomaly ratios are larger than 10%, we set the maximum anomaly ratio as 10%). For KDD-Rev, we set the maximum anomaly ratio to 2.5% because the performance of GOAD (without SRR) drops significantly even with a small ratio of anomalies in the training data. Fig. 4 shows that integrating SRR significantly improves GOAD (the state-of-the-art methods for tabular OCC). The improvements are more significant especially at higher anomaly ratios. This underlines how SRR can achieve significant improvements for both small (Thyroid & Arrhythmia) and large-scale datasets (KDD & KDD-Rev).

## 4.2 EXPERIMENTS ON IMAGE DATA

**Datasets.** We evaluate SRR on visual AD benchmarks, including CIFAR-10 [29], f-MNIST [56], Dog-vs-Cat [19], and MVTec [8]. For CIFAR-10, f-MNIST, and Dog-vs-Cat datasets, samples from

---

[1]Note that the experimental settings with contaminated training data in GOAD [7] and DAGMM [60] are slightly different from ours. Our contamination ratio is defined as the anomaly ratio over the entire training data, while their contamination ratio is the anomaly ratio over all the anomalies in the dataset.

one class are set to be normal and the rest from other classes are set to be an anomaly. Similar to the experiments on tabular data in Section. 4.1, we swap a certain amount of the normal training data with anomalies given the target anomaly ratio. For MVTec, since there are no anomalous data available for training, we borrow 10% of the anomalies from the test set and swap them with normal samples in the training set. Note that 10% of samples borrowed from the test set are excluded from evaluation. For all datasets, we experiment with varying anomaly ratios from 0% to 10%.

We use area under ROC curve (AUC) and average precision (AP) metrics to quantify the performance for visual AD (with scale 0-100). When computing AP, we set the minority class of the test set as label 1 and majority as label 0 (e.g., normal samples are set as label 1 for CIFAR-10 experiments as there are more anomaly samples that are from 9 classes in the test set). We run all experiments with 5 random seeds and report the average performance for each dataset across all classes. Per-class AUC and AP are reported in Appendix A.4.

**Models.** For semantic AD benchmarks, CIFAR-10, f-MNIST, and Dog-vs-Cat, we compare the SRR with two-stage OCCs [51] using various representation learning methods, such as distribution-augmented contrastive learning [51], rotation prediction [23, 28] and its improved version [51], and denoising autoencoder. For MVTec benchmarks, we use CutPaste [33] as a baseline and compare to its version with SRR integration. For both experiments, we use the ResNet-18 architecture, trained from random initialization, using the hyperparameters from [51] and [33]. The same model and hyperparameter configurations are used for SRR with $K = 5$ classifiers in the ensemble. We set $\gamma$ as twice the anomaly ratio of training data. For 0% anomaly ratio, we set $\gamma$ as 0.5. Finally, a Gaussian Density Estimator (GDE) on learned representations is used as OCC.

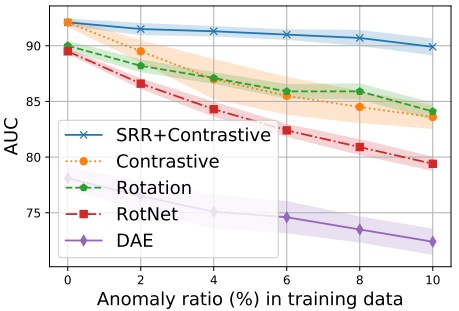 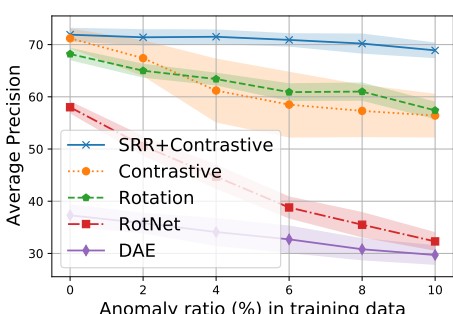

Figure 5: Unsupervised AD performance with various OCCs on CIFAR-10. For SRR we adapt distribution-augmented contrastive representation learning [51]. (Left) AUC, (Right) Average Precision (AP).

**Results.** Fig. 5 shows a significant performance drop with increased anomaly ratio, regardless of representation learning. For example, the AUC of distribution-augmented contrastive representation [51] drops from 92.1 to 83.6 when anomaly ratio becomes 10%. Similarly, the improved rotation prediction representation [28] drops from 90.0 to 84.1. On the other hand, SRR effectively handles the contamination in the training data and achieves 89.9 AUC with 10% anomaly ratio, reducing the performance drop by 74.1%. An 'oracle' upper bound would be the removal of all anomalies from the training data (which is the same as the performance at 0% anomaly ratio for the same size of the data). As Fig. 5 shows, the performance of SRR is similar to this oracle upper bound performance (less than 2.5 AUC difference) even with high anomaly ratios (10%). The results are also similar in other metrics, such as AP in Fig. 5 or Recall at Precision of 70 and 90 in Fig. 9 in the Appendix.

We repeat experiments on 3 additional visual AD datasets and report results in Fig. 6. We observe consistent and significant improvements over the baseline across different datasets and different one-class classification methods. Note that the improvement is more significant at higher anomaly ratios. For instance, on MVTec dataset, SRR improves AUC by 4.9 and AP by 7.1 compared to the state-of-the-art CutPaste OCC with an anomaly ratio of 10%. In the Appendix (Sec. A.4), we also illustrate per-class performance of SRR compared with the state-of-the-art.

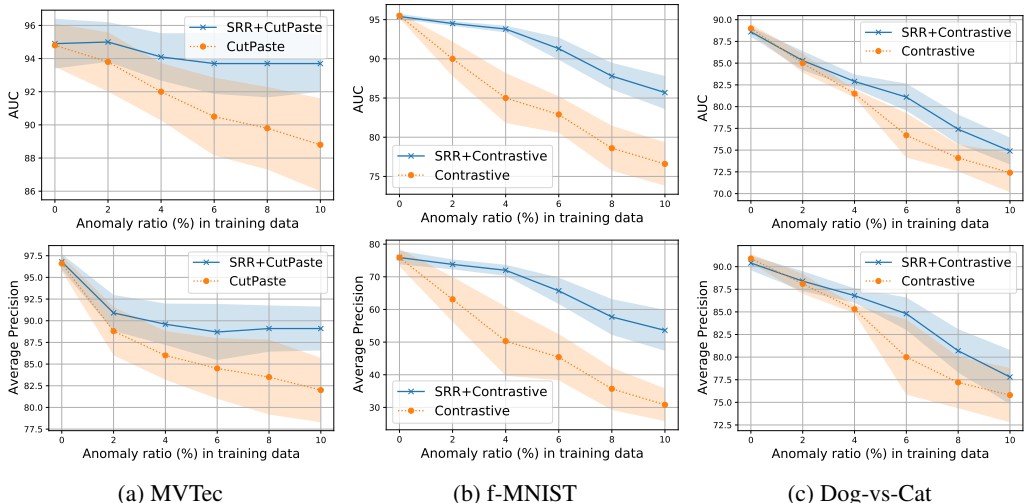

Figure 6: Unsupervised AD performance on (a) MVTec (b) f-MNIST, and (c) Dog-vs-Cat datasets with varying anomaly ratios. We use state-of-the-art one-class classification models for baselines, such as distribution-augmented contrastive representations [51] for f-MNIST and Dog-vs-Cat, or CutPaste [33] for MVTec, and build SRR on top of them.

### 4.3 PERFORMANCE ANALYSES

In this section, we conduct sensitivity analyses on two hyperparameters of SRR, namely, the number $K$ for ensemble OCCs and the percentile threshold $\gamma$ that determines normal and anomalies in the data refinement module. In addition, we show the importance of updating the representation with refined data. Ablation studies are conducted on two visual AD benchmarks, CIFAR-10 and MVTec. More experimental results and discussions can be found in the Appendix.

#### 4.3.1 SENSITIVITY TO HYPERPARAMETERS

SRR is designed for unsupervised AD and it is important to ensure robust performance against changes in the hyperparameters as model selection without labeled data would be very challenging. Fig. 7 presents the sensitivity analyses of SRR with respect to various hyperparameters. In Fig. 7a, we observe the performance improvement as we increase the number of classifiers for ensemble. This is particularly effective on CIFAR (top in Fig. 7a), where the number of samples in the training data is large enough that even with large $K$, the number of samples to train each OCC ($N/K$) would be sufficient. In Fig. 7b, we observe that SRR performs robustly when $\gamma$ is set to be larger than the actual anomaly ratio (10%). When $\gamma$ is less than 10%, however, we see a significant drop in performance, all the way to the baseline ($\gamma = 0$). Our results show that SRR improves upon baseline regardless of the threshold. In addition, it suggests that $\gamma$ could be set to be anywhere from the true anomaly ratio and and 2x the anomaly ratio to maximize its effectiveness. When the true anomaly ratio is unknown, as discussed in Appendix A.5, Otsu's method could be used.

#### 4.3.2 ITERATIVELY UPDATING REPRESENTATIONS WITH REFINED DATA

It is possible to decouple representation learning and data refinement of SRR, which would result in a three-stage framework, where we learn representations until convergence without data refinement, followed by the data refinement and learning OCC. Fig. 7c shows that SRR using a pre-trained then fixed representation (i.e. when data refinement is used for OCC only) already improves the performance upon the baseline with no data refinement at any stage of learning representation or classifier. Improvements on CIFAR-10 are 4.9 in AUC and 8.0 in AP; and on MVTec are 1.3 in AD and in 1.3 in AP. This underlines the effectiveness of our data refinement module. More results on applying SRR on raw tabular features or learned (and fixed) visual representations are provided in Appendix A.2. Moreover, we find that learning representation with refined data plays a crucial role, resulting in another improvement in AUC by 1.4, AP by 4.5 on CIFAR-10, and AUC by 3.2, AP by 4.8 on MVTec, upon a SRR using a fixed representation.

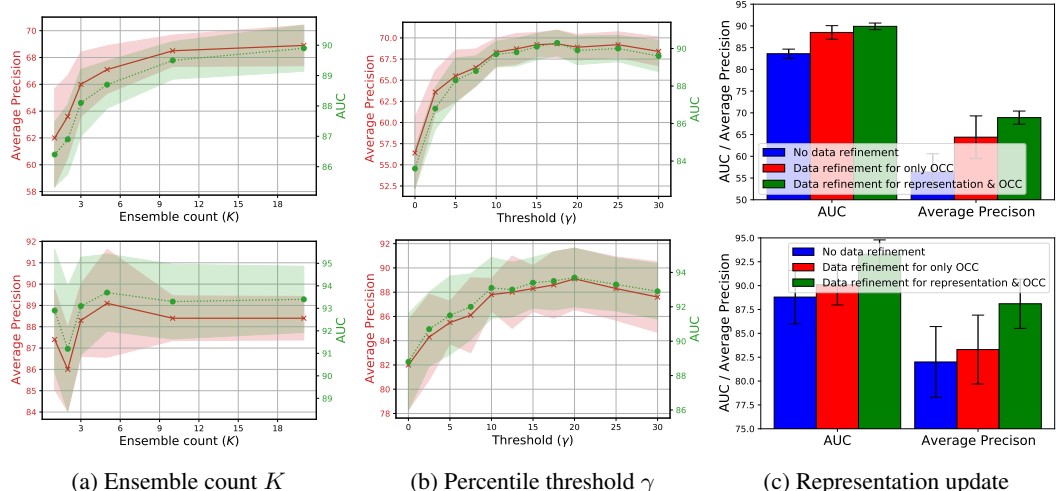

(a) Ensemble count $K$    (b) Percentile threshold $\gamma$    (c) Representation update

Figure 7: Ablation studies on (top) CIFAR-10 and (bottom) MVTec under 10% anomaly ratio setting with respect to (a) ensemble count $K$, (b) percentile threshold $\gamma$, and (c) data refinement with or without representation update.

## 4.4 QUANTIFYING THE REFINEMENT EFFICACY

We evaluate how many normal and anomalies are excluded by the proposed data refinement block. As in Fig. 8(a, b), with data refinement, we can exclude more than 80% of anomalies in the training set without removing too many normal samples. For instance, among 4% anomalies in CIFAR-10 data, SRR is able to exclude 80% anomalies while removing less than 20% normal samples. Such a high recall of anomalies of SRR is not only useful for unsupervised AD, but also could be useful for improving the annotation efficiency when a budget for active learning is available. Fig. 8(c, d) demonstrates the removed normal and abnormal samples by the data refinement module over training epochs. It shows that better representation learning (as training epochs increase) consistently improves the efficacy of the data refinement.

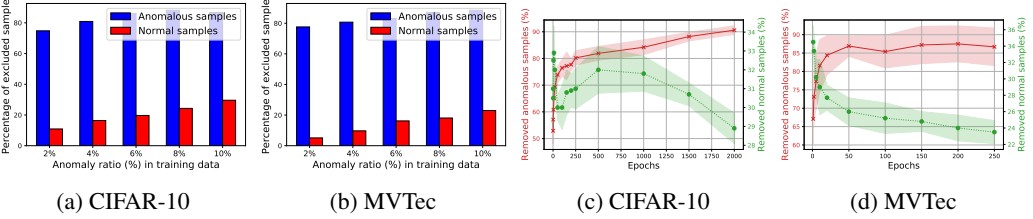

(a) CIFAR-10    (b) MVTec    (c) CIFAR-10    (d) MVTec

Figure 8: Percentage of excluded anomalous and normal samples by data refinement (a, b) with different anomaly ratios in training data and (c, d) over training epochs for 10% anomaly ratio.

## 5 CONCLUSION

AD has wide range of practical use cases. A challenging and costly aspect of building an AD system is that anomalies are rare and not easy to detect by humans, making them difficult to label. To this end, we propose a novel AD framework to enable high performance AD without any labels called SRR. SRR can be flexibly integrated with any OCC, and applied on raw data or on a trainable representations. SRR employs an ensemble of multiple OCCs to propose candidate anomaly samples that are refined from training, which allows more robust fitting of the anomaly decision boundaries as well as better learning of data representations. We demonstrate the state-of-the-art AD performance of SRR on multiple tabular and image data.

## ETHICS STATEMENT

SRR has the potential to make significant positive impact in real-world AD applications where detecting anomalies is crucial, such as for financial crime elimination, cybersecurity advances, or improving manufacturing quality. We note a potential risk associated with using SRR: when representations are not sufficiently good, there will be a negative cycle of refinement and representation updates/OCC. While we rely on the existence of good representations, in some applications they may be difficult to obtain (e.g., cybersecurity). This paper focuses on the unsupervised setting and demonstrates strong AD performance, opening new horizons for human-in-the-loop AD systems that are low cost and robust. We leave these explorations to future work.

## REPRODUCIBILITY STATEMENT

The source code of SRR will be published upon acceptance. For reproducibility, we only use public tabular and image datasets to evaluate the performances of SRR. Complete descriptions of those datasets can be found in **Datasets** subsections in Sec. 4.1 and 4.2. Detailed experimental settings and hyperparameters can be found in **Models** subsections in Sec. 4.1 and 4.2. The implementation details (including hyperparameters that we used) on GOAD [7] are described in Sec. A.3.

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

# A   APPENDIX

## A.1   ADDITIONAL RESULTS WITH DIFFERENT METRICS

There are various performance metrics of OCCs. In the main manuscript, we mainly use AUC, Average Precision (AP) and F1-score as the evaluation metrics. In this subsection, we report the performances of the proposed model (SRR) and baselines in terms on Recall at Precision 70 and 90 as the additional metrics on CIFAR-10 dataset.

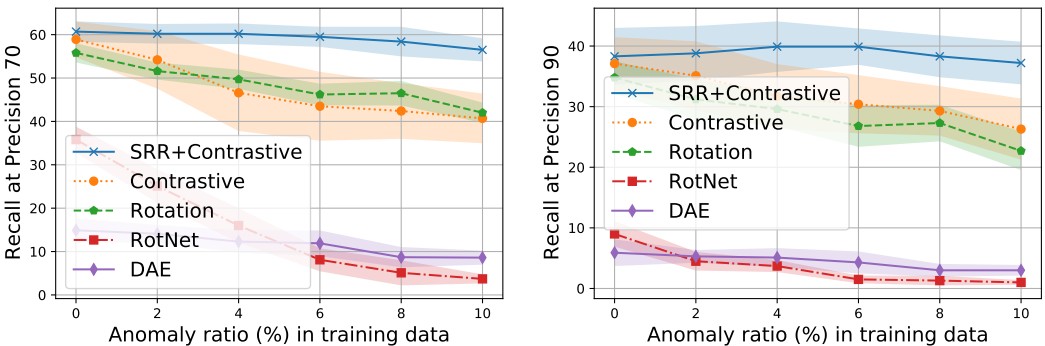

Figure 9: Performance of various OCCs on CIFAR-10 dataset. SRR is applied on top of Contrastive [51]. (Left) Recall at Precision 70, (Right) Recall at Precision 90.

As in Fig. 9, we observe a similar trends with these two additional metrics as well. For example, the performances of SRR are robust across various anomaly ratios. On the other hand, all the other OCCs show consistent and significant performance degradation as the anomaly ratio increases. SRR performs 15.8 and 10.9 better than the state-of-the-art OCC [51] in terms of recall at precision 70 and 90, respectively.

## A.2   SRR ON RAW TABULAR FEATURES / LEARNED IMAGE REPRESENTATIONS

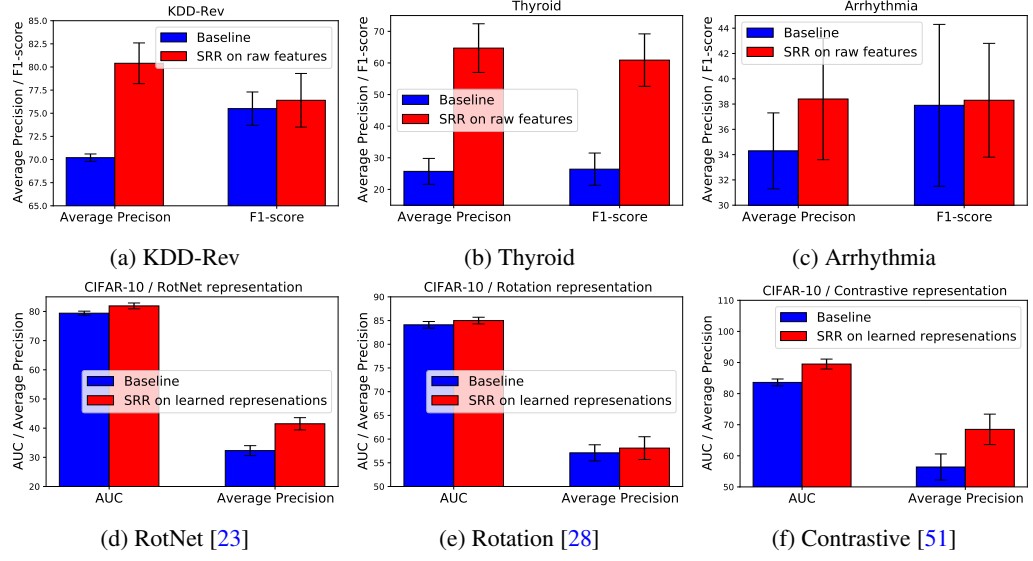

Figure 10: Performance of SRR on (top) raw tabular features and (lower) learned image representations. SRR consistently outperforms baseline and in some cases (e.g., Thyroid, and Contrastive [51]), the performance improvements are significant.

SRR is also applicable on raw tabular features or learned image representation without representation update using data refinement. In this section, we demonstrate the performance improvements by SRR without representation update to verify the effectiveness of data refinement block of SRR for shallow OCCs.

Fig. 10 (upper) demonstrates consistent and significant performance improvements when we apply SRR on top of raw tabular features. Specifically, the Average Precision (AP) improvements are 10.2, 29.0, and 4.1 with KDD-Rev, Thyroid, and Arrhythmia tabular datasets, respectively.

We also apply SRR on top of various learned image representations. As can be seen in Fig. 10 (lower), the performance improvements of SRR are consistent across various different learned image representations (without representation update). For instance, the AP improvements are 9.2, 1.0, and 12.1 with learned image representations using RotNet [23], Rotation [28], and Contrastive [51], respectively.

### A.3 Implementation Details on GOAD [7] for Tabular Data Experiments

A classification-based AD method, GOAD [7], has demonstrated strong AD performance on tabular datasets. Unlike previous works [23, 26] that formulate a parametric classifier for multiple transformation classification, GOAD employs distance-based classification of multiple transformations. For the set of transformations $T_m : \mathcal{X} \to \mathcal{D}$, $m = 1, ..., M$, the loss function of GOAD is written as in equation 3 with the probability defined in equation 4.

$$\mathcal{L} = -\mathbb{E}_{m,x}\big[\log P(m|T_m(x))\big], \tag{3}$$

$$P(\hat{m}|T_m(x)) = \frac{\exp(-\|f(T_m(x)) - c_{\hat{m}}\|^2)}{\sum_n \exp(-\|f(T_m(x)) - c_n\|^2)}, \tag{4}$$

where the centers $c_m$'s are updated by the average feature over the training set. While it is shown to perform well [7], we find that the distance-based formulation is not necessary, and we achieve the similar performance, if not worse, to [7] using a parametric classifier when computing the probability:

$$P(\hat{m}|T_m(x)) = \frac{\exp\big(w_{\hat{m}}^\top f(T_m(x)) + b_{\hat{m}}\big)}{\sum_n \exp\big(w_n^\top f(T_m(x)) + b_n\big)} \tag{5}$$

The formulation in equation 5 is easier to optimize than its original form in equation 4 as it can be fully optimized with backpropagation without alternating updates of feature extractor $f$ and centers $c_m$. Once we learn a representation by optimizing the loss in equation 3 using equation 5, we follow a two-stage one-class classification framework of [51] to construct a set of Gaussian density estimation OCCs for each transformation. Finally, we aggregate a maximum normality scores from a set of classifiers as the normality score.

In Table 1, we summarize the implementation details, such as network architecture or hyperparameters, and AD performance under clean training data setting that reproduces the results in [7].

| Datasets | KDD | KDD-Rev | Thyroid | Arrhythmia |
|---|---|---|---|---|
| F-score [7] | $98.4_{\pm 0.2}$ | $98.9_{\pm 0.3}$ | $74.5_{\pm 1.1}$ | $52.0_{\pm 2.3}$ |
| F-score (ours) | $98.0_{\pm 0.2}$ | $95.0_{\pm 0.2}$ | $75.1_{\pm 2.4}$ | $54.8_{\pm 3.2}$ |
| $f$ (feature) | $\big[$Linear(8), LeakyReLU(0.2)$\big]$ $\times 5$ | | | |
| Optimizer | Momentum SGD (momentum$= 0.9$) | | | |
| Learning rate | 0.001 | | | |
| L2 weight regularization | 0.00003 | | | |
| Batch size | $64 \times M$ | | | |
| Random projection dimension | 32 | | | |
| $M$ (number of transformations) | 32 | | 256 | |
| Train steps | $2^{10}$ | | $2^{16}$ | |

Table 1: The AD performance under clean only data setting of GOAD in [7] and our implementation. Our implementation demonstrates comparable, if not worse, performance to those reported in [7]. Our implementation also shares most hyperparameters across datasets except the $M$, the number of transformations, and the train steps, which are closely related to the size of training data.

## A.4 PER-CLASS AUC AND AP

In the main manuscript, we report the mean and standard deviation of AUC and AP across all classes in each dataset. In this section, we report the mean and standard deviation of AUC and AP for each class in each dataset (including CIFAR-10 (Table 2), MVTec (Table 3), fMNIST (Table 4), and Dog-vs-Cat (Table 5) datasets).

### A.4.1 PER-CLASS AUC AND AP ON CIFAR-10 DATASET

| anomaly ratios | 0% | | 10% | |
|---|---|---|---|---|
| Classes / Methods | Baseline [51] | SRR + [51] | Baseline [51] | SRR + [51] |
| AUC | | | | |
| Airplane | $90.5_{\pm0.6}$ | $90.4_{\pm0.4}$ | $85.3_{\pm0.8}$ | $87.6_{\pm0.7}$ |
| Automobile | $98.8_{\pm0.1}$ | $98.7_{\pm0.1}$ | $95.2_{\pm0.5}$ | $97.9_{\pm0.3}$ |
| Bird | $87.5_{\pm0.6}$ | $87.1_{\pm0.8}$ | $75.6_{\pm1.1}$ | $84.7_{\pm1.2}$ |
| Cat | $81.5_{\pm0.5}$ | $80.3_{\pm0.7}$ | $68.9_{\pm1.0}$ | $78.5_{\pm1.5}$ |
| Deer | $90.3_{\pm0.4}$ | $90.2_{\pm0.5}$ | $79.9_{\pm0.8}$ | $87.4_{\pm0.5}$ |
| Dog | $90.8_{\pm0.7}$ | $90.8_{\pm0.6}$ | $79.2_{\pm2.5}$ | $89.6_{\pm0.6}$ |
| Frog | $92.0_{\pm0.6}$ | $91.4_{\pm0.9}$ | $78.5_{\pm2.1}$ | $87.5_{\pm1.3}$ |
| Horse | $97.8_{\pm0.1}$ | $97.9_{\pm0.1}$ | $93.3_{\pm0.5}$ | $96.0_{\pm0.6}$ |
| Ship | $96.5_{\pm0.2}$ | $96.4_{\pm0.1}$ | $91.5_{\pm0.9}$ | $95.2_{\pm0.5}$ |
| Truck | $95.1_{\pm0.3}$ | $95.8_{\pm0.1}$ | $88.9_{\pm0.4}$ | $94.7_{\pm0.4}$ |
| Average Precision | | | | |
| Airplane | $68.3_{\pm1.7}$ | $67.9_{\pm1.1}$ | $61.3_{\pm1.9}$ | $61.8_{\pm1.1}$ |
| Automobile | $93.9_{\pm0.2}$ | $94.1_{\pm0.1}$ | $85.9_{\pm3.4}$ | $91.5_{\pm0.7}$ |
| Bird | $62.2_{\pm2.1}$ | $61.7_{\pm2.4}$ | $42.4_{\pm3.0}$ | $57.5_{\pm2.3}$ |
| Cat | $44.0_{\pm2.4}$ | $42.9_{\pm1.9}$ | $30.3_{\pm3.8}$ | $44.9_{\pm0.9}$ |
| Deer | $59.5_{\pm0.8}$ | $58.8_{\pm1.2}$ | $42.9_{\pm3.5}$ | $53.6_{\pm1.5}$ |
| Dog | $59.1_{\pm6.5}$ | $59.8_{\pm4.7}$ | $39.7_{\pm9.1}$ | $64.5_{\pm2.0}$ |
| Frog | $72.7_{\pm2.0}$ | $70.4_{\pm6.9}$ | $38.5_{\pm11.0}$ | $73.6_{\pm3.6}$ |
| Horse | $89.0_{\pm0.7}$ | $89.2_{\pm0.6}$ | $80.6_{\pm1.9}$ | $86.2_{\pm1.1}$ |
| Ship | $83.9_{\pm0.8}$ | $83.7_{\pm0.6}$ | $74.9_{\pm2.4}$ | $79.8_{\pm0.7}$ |
| Truck | $79.0_{\pm1.6}$ | $81.3_{\pm0.4}$ | $67.0_{\pm1.9}$ | $75.8_{\pm1.2}$ |

Table 2: The AD performance per class on CIFAR-10 dataset in terms of AUC and AP. Class represents the label of normal samples.

### A.4.2 PER-CLASS AUC AND AP ON MVTEC DATASET

| anomaly ratios | 0% | | 10% | |
|---|---|---|---|---|
| Classes / Methods | Baseline [33] | SRR + [33] | Baseline [33] | SRR + [33] |
| AUC | | | | |
| Bottle | $98.0_{\pm1.0}$ | $98.8_{\pm0.7}$ | $94.3_{\pm0.8}$ | $98.7_{\pm0.7}$ |
| Cable | $77.7_{\pm0.7}$ | $77.9_{\pm2.6}$ | $72.9_{\pm3.3}$ | $80.4_{\pm0.9}$ |
| Capsule | $96.1_{\pm1.1}$ | $97.2_{\pm1.3}$ | $92.4_{\pm1.7}$ | $96.8_{\pm0.7}$ |
| Hazelnut | $97.2_{\pm0.7}$ | $97.5_{\pm0.4}$ | $83.5_{\pm3.6}$ | $96.7_{\pm0.6}$ |
| Metal Nut | $98.5_{\pm1.0}$ | $98.6_{\pm0.4}$ | $91.2_{\pm2.8}$ | $98.7_{\pm0.3}$ |
| Pill | $92.1_{\pm3.1}$ | $91.1_{\pm2.5}$ | $85.1_{\pm1.5}$ | $92.1_{\pm0.5}$ |
| Screw | $87.4_{\pm1.6}$ | $86.8_{\pm2.5}$ | $79.9_{\pm1.0}$ | $86.4_{\pm1.4}$ |
| Toothbrush | $98.5_{\pm1.5}$ | $98.3_{\pm1.1}$ | $98.6_{\pm1.3}$ | $99.3_{\pm1.2}$ |
| Transistor | $96.4_{\pm2.0}$ | $96.7_{\pm1.6}$ | $93.9_{\pm2.8}$ | $95.7_{\pm0.7}$ |
| Zipper | $99.4_{\pm0.4}$ | $98.0_{\pm0.6}$ | $91.1_{\pm2.9}$ | $99.2_{\pm0.7}$ |
| Carpet | $90.0_{\pm3.9}$ | $90.8_{\pm4.6}$ | $80.4_{\pm6.1}$ | $91.2_{\pm2.7}$ |
| Grid | $99.2_{\pm0.5}$ | $99.7_{\pm0.3}$ | $92.9_{\pm7.6}$ | $99.7_{\pm0.2}$ |
| Leather | $99.8_{\pm0.3}$ | $99.7_{\pm0.4}$ | $98.0_{\pm1.1}$ | $100.0_{\pm0.0}$ |
| Tile | $93.1_{\pm1.5}$ | $93.3_{\pm2.7}$ | $86.4_{\pm3.4}$ | $92.4_{\pm1.8}$ |
| Wood | $98.8_{\pm0.9}$ | $98.5_{\pm1.4}$ | $92.1_{\pm2.2}$ | $99.0_{\pm0.5}$ |
| Average Precision | | | | |
| Bottle | $98.8_{\pm0.7}$ | $99.3_{\pm0.5}$ | $94.3_{\pm1.6}$ | $98.0_{\pm0.6}$ |
| Cable | $81.8_{\pm0.9}$ | $82.3_{\pm1.7}$ | $70.8_{\pm4.3}$ | $74.1_{\pm3.0}$ |
| Capsule | $99.0_{\pm0.3}$ | $99.3_{\pm0.3}$ | $79.5_{\pm5.3}$ | $89.8_{\pm2.7}$ |
| Hazelnut | $97.1_{\pm0.6}$ | $97.0_{\pm0.4}$ | $88.8_{\pm3.6}$ | $93.6_{\pm1.9}$ |
| Metal Nut | $99.6_{\pm0.3}$ | $99.6_{\pm0.1}$ | $76.1_{\pm8.4}$ | $92.1_{\pm3.1}$ |
| Pill | $98.2_{\pm0.7}$ | $97.9_{\pm0.6}$ | $56.6_{\pm2.6}$ | $67.3_{\pm2.2}$ |
| Screw | $93.8_{\pm1.1}$ | $93.6_{\pm1.3}$ | $60.1_{\pm2.8}$ | $63.7_{\pm1.8}$ |
| Toothbrush | $99.2_{\pm0.9}$ | $99.2_{\pm0.6}$ | $97.4_{\pm2.2}$ | $98.6_{\pm2.0}$ |
| Transistor | $93.0_{\pm3.0}$ | $93.7_{\pm2.4}$ | $97.8_{\pm1.6}$ | $97.1_{\pm0.5}$ |
| Zipper | $99.8_{\pm0.2}$ | $99.2_{\pm0.3}$ | $86.1_{\pm2.8}$ | $98.0_{\pm1.5}$ |
| Carpet | $94.4_{\pm3.5}$ | $95.7_{\pm2.8}$ | $64.2_{\pm8.2}$ | $71.6_{\pm4.2}$ |
| Grid | $99.5_{\pm0.3}$ | $99.8_{\pm0.2}$ | $90.9_{\pm6.1}$ | $99.4_{\pm0.4}$ |
| Leather | $99.9_{\pm0.2}$ | $99.9_{\pm0.2}$ | $97.2_{\pm1.3}$ | $100.0_{\pm0.1}$ |
| Tile | $96.1_{\pm0.9}$ | $96.4_{\pm1.4}$ | $82.4_{\pm4.5}$ | $86.4_{\pm2.8}$ |
| Wood | $99.4_{\pm0.5}$ | $99.4_{\pm0.6}$ | $88.1_{\pm3.2}$ | $97.2_{\pm1.7}$ |

Table 3: The AD performance per class on MVTec dataset in terms of AUC and AP. Class represents the label of normal samples.

### A.4.3 PER-CLASS AUC AND AP ON FMNIST DATASET

### A.4.4 PER-CLASS AUC AND AP ON DOG-VS-CAT DATASET

| anomaly ratios | 0% | | 10% | |
|---|---|---|---|---|
| Classes / Methods | Baseline [51] | SRR + [51] | Baseline [51] | SRR + [51] |
| AUC | | | | |
| T-shirt/top | $93.3_{\pm1.2}$ | $93.2_{\pm1.0}$ | $79.2_{\pm1.3}$ | $84.7_{\pm1.0}$ |
| Trouser | $99.2_{\pm0.3}$ | $99.2_{\pm0.2}$ | $92.9_{\pm0.8}$ | $94.5_{\pm1.9}$ |
| Pullover | $93.8_{\pm0.2}$ | $93.8_{\pm0.2}$ | $78.4_{\pm2.7}$ | $90.1_{\pm1.1}$ |
| Dress | $94.4_{\pm0.5}$ | $94.0_{\pm0.5}$ | $78.4_{\pm3.5}$ | $85.8_{\pm1.6}$ |
| Coat | $95.1_{\pm0.1}$ | $94.9_{\pm0.2}$ | $80.4_{\pm3.1}$ | $88.7_{\pm1.7}$ |
| Sandal | $95.4_{\pm0.3}$ | $95.5_{\pm0.4}$ | $67.6_{\pm4.3}$ | $81.7_{\pm1.8}$ |
| Shirt | $87.9_{\pm0.2}$ | $87.9_{\pm0.2}$ | $73.9_{\pm1.2}$ | $81.0_{\pm2.5}$ |
| Sneaker | $99.1_{\pm0.0}$ | $99.2_{\pm0.1}$ | $80.2_{\pm4.9}$ | $94.7_{\pm3.6}$ |
| Bag | $97.8_{\pm0.2}$ | $97.6_{\pm0.3}$ | $58.2_{\pm3.3}$ | $68.6_{\pm2.3}$ |
| Ankle boot | $98.6_{\pm0.2}$ | $98.5_{\pm0.2}$ | $76.6_{\pm2.6}$ | $87.1_{\pm3.6}$ |
| Average Precision | | | | |
| T-shirt/top | $68.7_{\pm12.1}$ | $67.9_{\pm8.9}$ | $43.0_{\pm4.7}$ | $56.8_{\pm3.7}$ |
| Trouser | $96.5_{\pm2.9}$ | $98.2_{\pm0.3}$ | $78.2_{\pm5.2}$ | $87.1_{\pm3.7}$ |
| Pullover | $62.7_{\pm1.2}$ | $63.6_{\pm1.3}$ | $27.2_{\pm6.9}$ | $57.1_{\pm1.5}$ |
| Dress | $72.3_{\pm3.7}$ | $71.6_{\pm2.7}$ | $36.1_{\pm8.8}$ | $57.3_{\pm2.6}$ |
| Coat | $66.2_{\pm1.1}$ | $65.6_{\pm1.0}$ | $24.3_{\pm3.9}$ | $50.7_{\pm9.5}$ |
| Sandal | $70.9_{\pm2.8}$ | $70.8_{\pm2.4}$ | $17.8_{\pm5.9}$ | $40.1_{\pm5.4}$ |
| Shirt | $49.6_{\pm0.9}$ | $49.7_{\pm0.9}$ | $27.2_{\pm5.8}$ | $37.6_{\pm4.7}$ |
| Sneaker | $91.9_{\pm0.4}$ | $92.9_{\pm0.9}$ | $21.0_{\pm3.8}$ | $81.6_{\pm11.4}$ |
| Bag | $92.2_{\pm0.4}$ | $91.8_{\pm1.0}$ | $13.2_{\pm2.0}$ | $20.1_{\pm3.7}$ |
| Ankle boot | $87.9_{\pm1.4}$ | $86.5_{\pm1.5}$ | $19.6_{\pm3.1}$ | $48.0_{\pm16.5}$ |

Table 4: The AD performance per class on fMNIST dataset in terms of AUC and AP. Class represents the label of normal samples.

| anomaly ratios | 0% | | 10% | |
|---|---|---|---|---|
| Classes / Methods | Baseline [51] | SRR + [51] | Baseline [51] | SRR + [51] |
| AUC | | | | |
| Dog | $89.7_{\pm0.6}$ | $89.2_{\pm0.4}$ | $74.7_{\pm2.1}$ | $79.1_{\pm1.3}$ |
| Cat | $88.2_{\pm0.4}$ | $88.0_{\pm0.7}$ | $70.1_{\pm2.4}$ | $70.7_{\pm1.8}$ |
| Average Precision | | | | |
| Dog | $91.6_{\pm0.7}$ | $91.0_{\pm0.6}$ | $80.0_{\pm2.2}$ | $84.0_{\pm1.2}$ |
| Cat | $90.2_{\pm0.3}$ | $89.8_{\pm1.0}$ | $71.6_{\pm3.8}$ | $71.5_{\pm4.8}$ |

Table 5: The AD performance per class on Dog-vs-Cat dataset in terms of AUC and AP. Class represents the label of normal samples.

## A.5 SRR with Otsu's method

In the main manuscript we show that the performances of the proposed unsupervised anomaly detection framework (SRR) are significantly better than the state-of-the-art OCCs if the given data is fully unlabeled. Although SRR does not need any labeled data, it needs a true anomaly ratio for hyper-parameter optimization. In this subsection, we further extend SRR to the setting where there is no access to the true anomaly ratio as well. We utilize the Otsu's method [49] to identify the threshold between normal and anomalous samples and use that as the hyper-parameter ($\gamma$) instead of twice of the true anomaly ratio. Note that other previous works [5, 42, 55] also utilized intra-variance minimization, in a similar vein to Otsu's method, for determining the threshold for data refinement.

The key idea of the Otsu's method is to find the threshold that minimizes the intra-class variance. This is defined as the weighted sum of variances of the two classes. Let us denote the normality scores as $\{s_i\}_{i=1}^N$ and threshold as $\eta$. Then, we try to find the threshold ($\eta$) that minimizes the weighted sum of the variance $(w_0(\eta) \times \sigma_0(\eta) + w_1(\eta) \times \sigma_1(\eta))$ where $w_0(\eta) = \sum_{i=1}^N \mathbb{I}(s_i < \eta)/N$ and $w_1(\eta) = \sum_{i=1}^N \mathbb{I}(s_i \geq \eta)/N$. $\sigma_0(\eta)$ and $\sigma_1(\eta)$ are the variances of each class. The optimal threshold ($\eta^*$) is determined as

$$\eta^* = \min_\eta w_0(\eta) \times \sigma_0(\eta) + w_1(\eta) \times \sigma_1(\eta).$$

We use the twice of $\eta^*$ as the hyperparameter ($\gamma$) in SRR.

We evaluate the performances of Otsu's method (on top of SRR) in comparison to the state-of-the-art OCC [33] and original SRR with the knowledge of the true anomaly ratio using MVTec dataset.

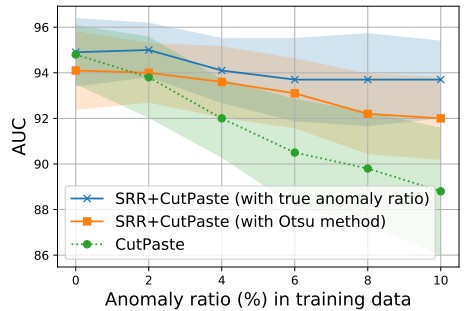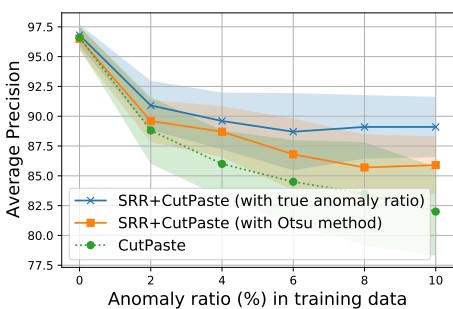

Figure 11: Unsupervised anomaly detection performances with Otsu's method on top of SRR on MVTec dataset. (Left) AUC, (Right) Average Precision (AP).

Fig. 11 demonstrates that even without true anomaly ratio, the performance of SRR can be significantly better than the state-of-the-art OCC [33] with Otsu's method. Also, Fig. 11 shows that the knowledge of true anomaly ratio is crucial information for maximizing the performance of SRR in fully unsupervised settings.

We further extend the experimental results using Otsu's method with other datasets such as CIFAR-10 and Thyroid datasets. Table 6 are the overall results - Otsu's method yields only slight degradation compared to SRR with true anomaly ratio; however, it still significantly outperforms SOTA OCC baselines.

| Datasets / Method | SOTA OCC | SRR with Otsu's method | SRR |
|---|---|---|---|
| CIFAR-10 | 0.855 / 0.585 | 0.906 / 0.703 | **0.910 / 0.709** |
| Thyroid | 0.506 | 0.623 | **0.639** |

Table 6: Additional ablation studies. SOTA OCC methods are [51] for CIFAR-10 dataset and GOAD [7] for Thyroid dataset. We introduce 6% noise on MVTec dataset and 1.5% noise on the Thyroid dataset. Metrics are (AUC/AP) for CIFAR-10 dataset and F1 score for Thyroid dataset.

## A.6 ADDITIONAL ABLATION STUDIES

To better understand the source of gains, we include comparisons between the final ensemble model on the converged self-supervised extractor (SRR without final OCC) with the proposed SRR (using an additional final OCC). Also, to further support the novelty of SRR, we report additional experimental results only with ensemble model without data refinement (Ensemble Only).

As can be seen in Table 7, the proposed version of SRR (with an additional final OCC) outperforms significantly, which is attributed to the fact that while fitting the individual OCC models in the ensemble, we do not exclude the possible anomaly samples for diversity of the trained submodels, so the anomaly decision boundaries can be fitted robustly. Therefore, the ensemble model can be somewhat less accurate for classifying difficult samples (near the decision boundary between normal

and abnormal) compared to the final OCC used (that is trained with only normal samples). Also, employment of ensemble learning without data refinement (Ensemble only), yields much worse performance than the proposed method (SRR), underlining the importance of the core data refinement idea of SRR.

| Datasets / Method | SOTA OCC | Ensemble Only | SRR without final OCC | SRR |
|---|---|---|---|---|
| MVTec | 0.905 / 0.845 | 0.911 / 0.849 | 0.922 / 0.870 | **0.937 / 0.887** |
| CIFAR-10 | 0.855 / 0.585 | 0.862 / 0.599 | 0.890 / 0.677 | **0.910 / 0.709** |

Table 7: Additional ablation studies. SOTA OCC methods are CutPaste [33] for MVTec dataset and [51] for CIFAR-10 dataset. We introduce 6% noise on both datasets. Metrics are (AUC/AP).

### A.7 ADDITIONAL BASELINES

We add extra baselines from robust anomaly detection literature: Standard PCA [54], Robust PCA [15] and Local Outlier Factor (LOF) [13] for tabular data and Robust autoencoder (Robust AE) [59] for image data. As can be seen in Table 8, the performance of PCA and LOF are highly degraded even with a small amount of anomalies in the training data. For Robust PCA and Robust AE, the performance degradation is less but still significant in comparison to SRR. Overall, SRR outperforms other benchmarks in fully unsupervised settings, underlining the importance of data refinement in improving the robustness to anomaly ratio in training data, as the core constituent of SRR framework.

| Datasets / Method | PCA | Robust PCA | LOF | Robust AE | SRR |
|---|---|---|---|---|---|
| CIFAR-10 | - | - | - | 0.636 / 0.174 | **0.910 / 0.709** |
| Thyroid | 0.299 | 0.377 | 0.338 | - | **0.506** |
| KDD | 0.836 | 0.893 | 0.873 | - | **0.942** |

Table 8: Additional experiments with extra baselines from robust anomaly detection literature. We introduce 6% noise on CIFAR-10 and KDD datasets. For Thyroid dataset, we introduce 1.5% noise. Metrics are (AUC/AP) for image data and F1 score for tabular data.

### A.8 CONVERGENCE GRAPHS

The proposed SRR framework is converged when the iterative training of the self-supervised learning is converged. Depending on the data refinement model training, corresponding self-supervised models are also trained with differently refined training data. Usually, the data refinement model and self-supervised models converge after a similar number of epochs. Fig. 12 illustrate the convergence graphs of SRR with MVTec and CIFAR-10 datasets.

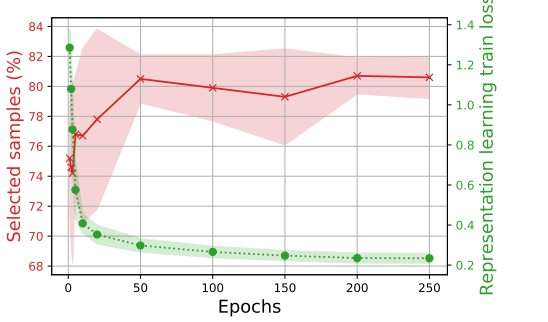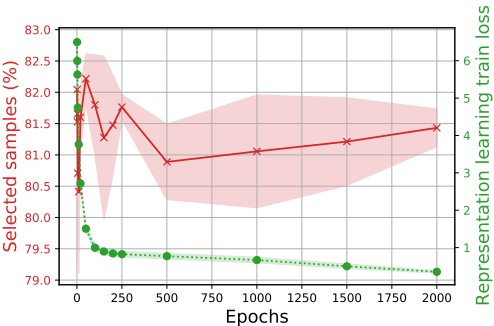

Figure 12: Convergence graphs of SRR with (left) MVTec dataset, (right) CIFAR-10 dataset.

### A.9 ADDITIONAL DISCUSSIONS: RELATED WORKS FOR DATA REFINEMENT

In this subsection, we further discuss the related works [5, 42, 55] for data refinement that we briefly mentioned in Section 2. We highlight the differences between SRR and [5, 42, 55] below.

First, [5, 55] are based on auto-encoders and they directly utilize the reconstruction errors as informative signals for anomaly detection and iterative data refinement. However, prior work [43] discovered that the reconstruction is not a good informative signal for outlier detection. Also as shown in Fig. 5, the performance of DAE (reconstruction errors based anomaly detection model) works much worse than alternatives.

Second, SRR utilizes ensemble learning for data refinement to improve the robustness which is critical in unsupervised anomaly detection. On the other hand, [5, 42, 55] utilize a single model for data refinement. As can be seen in Fig. 7a, the impact of the ensemble model for data refinement is significant.

Third, [5, 42, 55] directly utilize the (pseudo) abnormal samples for model training in addition to (pseudo) normal samples. For instance, the model in [42] is trained via two-class ordinal regression where two classes come from (pseudo) normal and (pseudo) abnormal samples. Directly relying on the (imperfectly) labeled abnormal samples can be harmful for the anomaly detection due to the overfitting problems and high False Positive Rates (FPR) in (pseudo) abnormal samples. For instance, with 80% of the recall (for the abnormal sample discovery), the precisions (for the abnormal sample discovery) are 28.2% for CIFAR-10 and 31.7% with MVTec datasets (with 6% abnormal ratio) using SRR framework. It means that the majority of the (pseudo) abnormal samples are actually the normal samples (71.8% for CIFAR-10 and 68.3% for MVTec datasets). This would be strong evidence that directly utilizing the (pseudo) abnormal samples for training can be harmful.

Lastly, [42, 55] are based on the end-to-end anomaly detection models which are empirically less accurate than two-stage models [51].

### A.10 ADDITIONAL DISCUSSIONS: COMPUTATIONAL COMPLEXITY

Note that when applying SRR on top of representation learning, the computational complexity of the representation learning part is not changed. With SRR, the additional computations come from training the ensemble models on top of learned representations. Note that we use shallow one-class classifiers (such as GDE or one-class SVM) for submodels; thus, the additional computational complexity is marginal. We would like to mention that all the experiments are done on a single V100 GPU and each experiment needs at most 12 hours for training (the additional training time caused by the SRR framework is an average 13.1% of the total training time). The computations of the ensemble parts can be further improved by the model parallelization.

