# OpenReview forum: "Self-Supervise, Refine, Repeat: Improving Unsupervised Anomaly Detection"
_ICLR.cc/2022/Conference — ICLR 2022 Submitted_

### Official Review · Reviewer_s1MH · 2021-11-01

**Correctness:** 4
**Technical Novelty And Significance:** 3
**Empirical Novelty And Significance:** 3
**Recommendation:** 6
**Confidence:** 5

**Main Review:**

The authors propose a novel AD framework to enable inspect defects with one-class, which is called SRR and applicable on unlabeled datasets.SRR employs an ensemble of multiple OCCs to give the potential anomaly refined samples from training. SRR brings the advantages of making the anomaly decision boundaries more robust and giving better data representations. The proof and experiment results are well organized.

**Summary Of The Paper:**

The authors propose a data refinement approach combined with self-supervised representation to robust one-class classification, which is commonly used in the anomaly detection scenario. The proposed data refinement approach is designed based on an ensemble of one-class classifiers.

**Summary Of The Review:**

The paper is ready for acceptance.

---

> ### Author Response · Authors · 2021-11-20
> **Answers to the reviewer s1MH's reviews**
>
> [R3-A1]: Thank you for all your positive comments about our paper and finding it ready for acceptance.

---

### Official Review · Reviewer_gVix · 2021-11-05

**Correctness:** 4
**Technical Novelty And Significance:** 3
**Empirical Novelty And Significance:** 2
**Recommendation:** 6
**Confidence:** 5

**Main Review:**

[Stregnth]
The effectiveness of the proposed framework is validated on top of contrastive learning-based models, which are state-of-the-arts.
Extensive experiments on both tabular and image datasets support the effectiveness of the framework.
Ablation studies decouple the effects of each hyperparameter. Also, the representation update study shows that re-training representation with the refined dataset is important.

[Weakness]
- Although the proposed framework is tested on contrastive models, the idea of data refinement itself is independent of these models.
The framework can be applied to other types of anomaly detectors but it is not shown.

- The iterative refinement has been studied [1,2,3] previously, however, no comparison or discussion is addressed with those approaches.
Although they use AE-based models, the idea of iterative refinement can be deployed on top of contrastive models as well. What makes SRR more effective than these methods? What is the main factor that makes SRR more competitive than these methods?

[1] Xia, Yan, et al. "Learning discriminative reconstructions for unsupervised outlier removal." Proceedings of the IEEE International Conference on Computer Vision. 2015.

[2] Beggel, Laura, Michael Pfeiffer, and Bernd Bischl. "Robust anomaly detection in images using adversarial autoencoders." arXiv preprint arXiv:1901.06355 (2019).

[3] Pang, Guansong, et al. "Self-trained deep ordinal regression for end-to-end video anomaly detection." Proceedings of the IEEE/CVF Conference on Computer Vision and Pattern Recognition. 2020.

- Hyperparameter selection requires tuning. Although it is shown in Figure 7, that any value of gamma improves over baseline, it is still important how to choose this value. The paper suggested Otsu's method as a solution to predict the approximate anomaly ratio of the dataset but it is not shown in the main experiments. (MVTec experiment is provided in A.5) In the main experiments, hyperparameter gamma is tuned with two times of anomaly ratio which this setting requires prior knowledge on anomaly ratio. I wonder how effective Otsu's method is in other scenarios.

- The training requires heavy computation as the framework requires ensemble learning on top of contrastive learning.

[Questions]
- What is the convergence condition? Is it necessary to train with the framework until the data refinement gives marginal change to the data? How long does it take?

- What is the important difference of SRR from the previous refinement methods? Is SRR more effective than the previous refinement approaches?

- How does SRR perform when trained with Otsu's method rather than using true anomaly ratio?


[Post rebuttal]
The authors addressed all my concerns.
I raise the score, assuming the rebuttal materials will be included in the revised version.
Rebuttal materials here means,
- OTsu's method experiment
- detailed discussion on the difference between SRR and the previous iterative works
- Convergence analysis (maybe this one in the appendix)

**Summary Of The Paper:**

The paper tackles an unsupervised anomaly detection problem where the training set contains an unknown portion of anomalies.
When anomalies are contained in the training set, it is known that classical AD approaches' performance degrades.
The idea is to filter out potential anomaly samples (data refinement) by ensemble model.
Each model in the ensemble is trained on a disjoint set of training data and then used as a classifier to determine potential anomalies.
Then the data refinement process uses a hard assignment excluding anomalies from the training set.
Refinement and ensemble training is repeated iteratively until convergence.
The proposed framework is validated on the four tabular datasets and four image datasets.

**Summary Of The Review:**

The paper proposes a framework to refine data and train contrastive models for unsupervised anomaly detection problems.
The extensive experiments show the effectiveness of the method.
However, the main experiments require prior knowledge of the true anomaly ratio which is unavailable in real-world problems.
Discussions about the important difference between the proposed method and the previous iterative methods would make the paper more convincing.

[Post rebuttal]
The authors addressed all my concerns.
I raise the score, assuming the rebuttal materials will be included in the revised version.
Rebuttal materials here means,
- OTsu's method experiment
- detailed discussion on the difference between SRR and the previous iterative works
- Convergence analysis (maybe this one in the appendix)

---

> ### Author Response · Authors · 2021-11-20
> **Answers to the reviewer gVix's questions [2/2]**
>
> [R2-A4: Computational complexity]:
> Note that when applying the SRR on top of contrastive learning, the computational complexity of the contrastive learning part is not changed. With SRR, the additional computations come from training the ensemble models on top of learned representations. Note that we use shallow one-class classifiers (such as GDE or one-class SVM) for submodels; thus, the additional computational complexity is marginal. Also, to further reduce the computational complexity, we update the data refinement ensembles only at certain epochs. More details can be found on the upper side of page 6 in the current manuscript. We would like to mention that all the experiments are done on a single V100 GPU and each experiment needs at most 12 hours for training (the additional training time caused by the SRR framework is an average 13.1% of the total training time). Note that the computations of the ensemble parts can be further improved by the model parallelization. We hope these clarify concerns about the computational requirements - we will discuss them further in the revised manuscript.
>
> [R2-A5: Convergence condition]:
> The proposed SRR framework is converged when the iterative training of the self-supervised learning is converged. Depending on the data refinement model training, corresponding self-supervised models are also trained with differently refined training data. Usually, the data refinement model and self-supervised models converge after a similar number of epochs. We will add the convergence graphs of the data refinement model and the self-supervised model in the revised appendix.

---

> > ### Comment · Reviewer_gVix · 2021-11-29
> > **The raised concerns are well adressed.**
> >
> > Thank you for the detailed one-by-one reply to the raised concerns.
> >
> > I agree with the other reviewers' opinions that the technical contributions are not significantly different from the previous works when each component (ensemble and iterative refinement) is compared individually.
> >
> > However, the critical concerns are well addressed during the rebuttal phase.
> > If the additional experiments and the discussions during the rebuttal phase are included in the revised version, I think the paper is qualified for the conference. I raise my score.

---

> ### Author Response · Authors · 2021-11-20
> **Answers to the reviewer gVix's questions [1/2]**
>
> [R2-A1: Generalizability of SRR to various anomaly detectors]:
> Thank you for finding all the aforementioned positive aspects in our work.
> Regarding your concern on the generality of data refinement beyond contrastive representation learning, actually, we already showed the outperformance of SRR with various different types of anomaly detectors in the current manuscript. On MVTec AD datasets (see Figure 6 (a)), we applied SRR on top of CutPaste [32]. For the tabular datasets, we applied SRR on top of GOAD [7]. Both representation learning methods, CutPaste and GOAD, are not based on contrastive learning. We hope these results underline the generalizability of SRR with any type of representation learning to be used with unsupervised anomaly detection.
>
> [R2-A2: Additional related works for data refinement]:
> Thank you for bringing these related works to our attention. Indeed we have already cited them in the manuscript and briefly discussed in Section 2 (Data refinement subsection). Here, we further highlight the differences between SRR and them.
> First, as mentioned, [1,2] are based on auto-encoders and they directly utilize the reconstruction errors as informative signals for anomaly detection and iterative data refinement. However, prior work [4] discovered that the reconstruction is not a good informative signal for outlier detection. Also as shown in Fig. 5 of the current manuscript, the performance of DAE (reconstruction errors based anomaly detection model) works much worse than alternatives.
> Second, SRR utilizes ensemble learning for data refinement to improve the robustness which is critical in unsupervised anomaly detection. On the other hand, [1,2,3] utilize a single model for data refinement. As can be seen in Fig. 7 (a), the impact of the ensemble model for data refinement is significant.
> Third, [1,2,3] directly utilize the (pseudo) abnormal samples for model training in addition to (pseudo) normal samples. For instance, the model in [3] is trained via two-class ordinal regression where two classes come from (pseudo) normal and (pseudo) abnormal samples. Directly relying on the (imperfectly) labeled abnormal samples can be harmful for the anomaly detection due to the overfitting problems and high False Positive Rates (FPR) in (pseudo) abnormal samples. For instance, with 80% of the recall (for the abnormal sample discovery), the precisions (for the abnormal sample discovery) are 28.2% for CIFAR-10 and 31.7% with MVTec datasets (with 6% abnormal ratio) using SRR framework. It means that the majority of the (pseudo-) abnormal samples are actually the normal samples (71.8% for CIFAR-10 and 68.3% for MVTec datasets). This would be strong evidence that directly utilizing the (pseudo-) abnormal samples for training can be harmful.
> Lastly, [1] and [3] are based on the end-to-end anomaly detection models which are empirically less accurate than two-stage models (Sohn et al. 2020).
>
> [4] Ren, Jie, et al. "Likelihood ratios for out-of-distribution detection." arXiv preprint arXiv:1906.02845 (2019).
>
> [R2-A3: Hyper-parameter Gamma optimization with Otsu’s method]:
> Thank you for the suggestion.
>
> Hyper-parameter optimization is a common problem for unsupervised anomaly detection in general. For SRR, one of the critical hyper-parameters is gamma. If we do not have the prior knowledge on the true anomaly ratio, we need to estimate it to determine gamma. There are multiple ways to identify the decision boundary between two distributions, and we utilize Otsu’s method which tries to minimize intra-class variance (which is the same as maximizing the inter-class variance). The previous works mentioned by the reviewers [1,2,3], utilized intra-variance minimization, in a similar vein to Otsu’s method, for determining the threshold for data refinement. Then, we use twice the estimated anomaly ratio as the gamma value for SRR. Note that setting the gamma as twice the (estimated) anomaly ratio is important for robust (pseudo) normal sample refinements. In the revised appendix, we will include the additional experimental results with other datasets such as CIFAR-10 and some tabular datasets with Otsu’s method.
> Below are the overall results - Otsu’s method yields only slight degradation compared to SRR with true anomaly ratio; however, it still significantly outperforms SOTA OCC baselines.
>
> Table 3: [Experimental details] SOTA OCC methods are CutPaste for MVTec dataset, Contrastive Learning for CIFAR-10, and GOAD for Thyroid in this experiment. We introduce 6% noise on both MVTec and CIFAR-10 datasets, and 1.5% noise on the Thyroid dataset. Metrics are (AUC/AP) or (F1).
>
> -------
> Datasets / Methods || SOTA OCC || SRR with Otsu's method || SRR with true anomaly ratio
>
> MVTec || 0.905 / 0.845 || 0.931 / 0.868 || 0.937 / 0.887
>
> CIFAR-10 || 0.855 / 0.585 || 0.906 / 0.703 || 0.910 / 0.709
>
> Thyroid || 0.506 || 0.623 || 0.639

---

### Official Review · Reviewer_7NNr · 2021-11-06

**Correctness:** 3
**Technical Novelty And Significance:** 2
**Empirical Novelty And Significance:** 4
**Recommendation:** 6
**Confidence:** 5

**Main Review:**

*Pros*
+ The experimental results demonstrate significant anomaly detection performance improvements for the proposed SRR approach, especially at high anomaly ratios.
+ The paper is overall presented and written well, as well as technically sound.
+ The paper is well placed well into the existing literature, up to including recent works.

*Cons*
- The methodological novelty of the proposed SRR approach is rather low (ensemble learning is standard to improve robustness and the individual detection method from Sohn et al. (2020) is not new)
- The experiments do not contain a comparison to any specifically robust AD approach (e.g. robust PCA for the tabular and robust autoencoders for the image dataset).
- While the paper includes ablation studies on key components and hyperparameters (ensemble size, data rejection confidence, updating the self-supervised feature extractor), I think there should also be a comparison to just using the final ensemble on the converged self-supervised extractor vs. training an additional final model. Is there much of a difference left?


Some additional minor points:

- I don't agree with the framing that most prior AD works "all depend on some labeled data" as expressed in the abstract and introduction. The bulk of AD research is on unsupervised methods (see Chandola et al. (2009) and the recent reviews by Ruff et al. (2021) and Pang et al. (2021)). However, I agree that most methods assume fairly clean training data. This view should be updated in my mind.
- The GDE abbreviation is used before it is explained.
- p.6: "For the N setting [...]" Typo?

**Summary Of The Paper:**

This paper proposes an ensemble approach, called SRR (Self-supervise, Refine, Repeat), for robust unsupervised anomaly detection. The proposed approach trains an ensemble of K detectors together with a joint self-supervised feature extractor g on K disjoint subsets of the data. The ensemble is then used to filter the training data, keeping only the data points that are collectively deemed normal by the K detectors. This training-data filtering process is repeated until the self-supervised feature extractor has converged. A final detector is then trained using the refined data and converged self-supervised feature extractor. Experiments on tabular and image datasets are presented which show that the proposed ensemble approach is more robust at high anomaly contamination ratios than respective state-of-the-art single detectors.

**Summary Of The Review:**

Though I think the methodological novelty of the proposed approach is rather low, and that the experimental comparison should be somewhat extended (where I expect the improvements of SRR to hold up), I am overall positive towards accepting this work since robust anomaly detection is a relevant problem of high practical significance, for which the proposed SRR approach demonstrates significant improvements over current state-of-the-art methods.

---

> ### Author Response · Authors · 2021-11-20
> **Answers to the reviewer 7NNr's questions**
>
> [R1-A1: Novelty]:
> Thank you for appreciating the aspects of our work on significant performance improvements, presenting and its place in the literature.
>
> Regarding novelty, our paper distinguishes by bringing a data-centric approach to unsupervised anomaly detection compared to the model-centric approaches in the literature. Essentially, our paper is the only one that shows significant unsupervised anomaly detection performance improvements by iterative employment of data refinement (with a selection mechanism based on an ensemble learning) along with self-supervised learning for better representations to employ anomaly detection on. We will rephrase the Introduction section to better express the novelty aspects.
> To further support the novelty claims above and address the comment regarding comparison to ensemble learning, we have added additional experiments, given in Table 1. As can be seen from the table below, employment of ensemble learning without data refinement (‘Ensemble only’), yields much worse performance than the proposed method (SRR), underlining the importance of the core data refinement idea of SRR.
>
> Table 1: [Experimental details] SOTA OCC methods are CutPaste for MVTec dataset and Contrastive Learning for CIFAR-10 in this experiment. We introduce 6% noise on both MVTec and CIFAR-10 datasets.
>
> Datasets / Methods (AUC/AP) || SOTA OCC || Ensemble Only || SRR without final OCC || SRR
>
> MVTec || 0.905 / 0.845 || 0.911 / 0.849 || 0.922 / 0.870 || 0.937 / 0.887
>
> CIFAR-10 || 0.855 / 0.585 || 0.862 / 0.599 || 0.890 / 0.677 || 0.910 / 0.709
>
>
> [R1-A2: Additional robust AD baselines]: Thank you for bringing these relevant baselines to our attention. We have added extra baselines from robust anomaly detection literature: Standard PCA, Robust PCA and Local Outlier Factor (LOF) for tabular data and Robust autoencoder for image data. Please see the experimental comparisons in Table 2. The performance of PCA and LOF are highly degraded even with a small amount of anomalies in the training data. For Robust PCA and Robust AE, the performance degradation is less but still significant in comparison to SRR. Overall, SRR outperforms other benchmarks in fully unsupervised settings, underlining the importance of data refinement in improving the robustness to anomaly ratio in training data, as the core constituent of SRR framework.
>
> Table 2: [Experimental details] We introduce 6% noise on CIFAR-10 and KDD datasets. For Thyroid, we introduce 1.5% noise. Metrics are (AUC/AP) or (F1).
>
> Datasets / Methods || PCA || Robust PCA || LoF || Robust AE || SRR
>
> CIFAR-10 || - || - || - || 0.636 / 0.174 || 0.910 / 0.709
>
> Thyroid || 0.299 || 0.377 || 0.338 || - || 0.506
>
> KDD || 0.836 || 0.893 || 0.873 || - || 0.942
>
>
> [R1-A3: Ablation study on final ensemble]:
> Thank you for the suggestion. To better understand the source of gains, we have added comparisons between the “final ensemble model” on the converged self-supervised extractor (SRR without final OCC) with the proposed SRR (using an additional final OCC). Please see the results in Table 1 in [R1-A1]. Overall, the proposed version of SRR (with an additional final OCC) outperforms significantly, which is attributed to the fact that while fitting the individual OCC models in the ensemble, we do not exclude the possible anomaly samples for diversity of the trained submodels, so the anomaly decision boundaries can be fitted robustly. Therefore, the ensemble model can be somewhat less accurate for classifying difficult samples (near the decision boundary between normal and abnormal) compared to the final OCC used (that is trained with only normal samples).
>
> [R1-A4: Clean data assumptions in the previous works]:
> Thanks for the note - we will rephrase our expression to frame this more precisely without overgeneralizing. Our major point was that most previous methods are designed with the assumption of clean training data. Constructing such clean training datasets necessitates excluding all the anomalies from the unlabeled samples, which indeed is a tedious labeling task. Yet, we will emphasize that some unsupervised AD methods like Deep SAD, DAGMM and GOAD have also considered settings with contaminated/anomalous data.
>
> [R1-A5: Abbreviation and typos]:
> GDE is the abbreviation of Gaussian Distribution Estimator- we had an explanation before using this abbreviation at the end of page 2. N setting is described in Figure 2(b) and actually it is not a typo - we will replace the term “N setting” with “Clean training data setting” to avoid the confusion.
>
> [R1-A6: Summary]:
> Thank you for the great suggestions and understanding the importance of the problem that we tackled in this paper. We have added some additional experiments including multiple new baselines and ablation studies with ensemble learning, as well as more justifications on the novelty of our paper. We hope these clarify all of your concerns and questions.

---

### Official Review · Reviewer_MspQ · 2021-11-19

**Correctness:** 3
**Technical Novelty And Significance:** 3
**Empirical Novelty And Significance:** 3
**Recommendation:** 5
**Confidence:** 4

**Main Review:**

**Strength**
1. the authors fully explore the power of unsupervised AD and shows outperformed results by using the proposed SRR scheme based on the GOAD framework.
2. The performance is improved by leveraging the ensemble of the OCC, which works well but may result in additional computational cost.
3. Enough details, such as the sensitivity of hyperparameters, are provided to reproduce the experiments on multiple tasks, including tabular and image datasets.

**Weakness**
1. The idea sounds promising but may not be the first work. The performance is enhanced by an ensemble of multiple tricks. It would be helpful to see a detailed ablation study, which may show more insights.
2. The baseline is mainly consisting of GOAD, OC-SVM, etc. Some recent SOTA baselines are missed, for example, NeuTral [1]
[1] Qiu, Chen, Timo Pfrommer, Marius Kloft, Stephan Mandt, and Maja Rudolph. "Neural Transformation Learning for Deep Anomaly Detection Beyond Images." ICML 2021.
3. The baseline varies case by case, for example, Cutpaste is used for MVTec datasets, have you considered the SOTA performance, for example in this link - https://paperswithcode.com/sota/anomaly-detection-on-mvtec-ad
Cupaste only ranks the 9th, have you compared with the other SOTA baselines?  I, therefore, have a concern, how do you choose the baseline method in different datasets? I suggest to choose more rather than a specific one.



**Summary Of The Paper:**

This paper proposes a self-supervised idea for unsupervised anomaly detection. Specifically, this framework enables high-performance AD without any labels via SRR, which is an ensemble approach to propose candidate anomaly samples that are refined from training. This way allows more robust fitting of the anomaly decision boundaries and also better learning of data representations.  Multiple examples are used to demonstrate the proposed methods on effectiveness and robustness.

**Summary Of The Review:**

Overall, the paper is well-written and the results and performance look solid. My major concern is the novelty, specifically compared with the GOAD method. In addition, the code is not uploaded, so I am not sure how it works for reproducibility and time cost.

---

> ### Author Response · Authors · 2021-11-20
> **Answers to the reviewer MspQ's questions**
>
> [R4-A1: Ablation study]:
> Thank you for your positive comments on the strengths of our method. Regarding novelty of our work, we note that our paper is the first one that shows significant unsupervised anomaly detection performance improvements by iterative employment of data refinement (with a selection mechanism based on an ensemble learning) along with self-supervised learning for better representations to employ anomaly detection on.
> To show further insights about the constituents of the performance, we expand our ablation studies. First, as can be seen in the additional experiments described in [R1-A1], we verify that the unsupervised learning performances do not improve only with the ensemble learning method. It underlines how much the data refinement is critical for improving the unsupervised anomaly detection performance. Also, in Figure 7(c), we show that it is important to apply data refinement for both representation learning and OCC models as another key contribution of our paper. Lastly, we show that the final OCC training is needed in addition to ensemble models in the additional experiments illustrated in [R1-A3]. Hopefully, these additional experiments and clarifications will help to understand the key contributors of SRR better. We will better clarify these in the Introduction and Methods sections.
>
> [R4-A2: Additional baselines for tabular data]:
> Thank you for bringing this work to our attention. We will cite and discuss this work. The major point we would like to emphasize is that SRR is a “data-centric” approach rather than “model-centric”, in other words, the key ideas of SRR can be combined with an unsupervised anomaly detection model (composed of representation learning and OCC). GOAD was the example we focused on for tabular data, and we showed SRR can be applied on top of GOAD to further improve its anomaly detection performance. Similarly, SRR can also be combined with NeuTral to further improve it. Especially for the regime of high anomaly ratio in the training data, the benefit of SRR can be significant. An important point to note is that the NeuTral paper only presents experiments for the setting where all the training data is normal (i.e. not truly “unsupervised” setting). When a truly “unsupervised” setting is considered, the deterioration of the contamination in the data can be mitigated by SRR. In conclusion, SRR would be complementary to NeuTral rather than competitor.
>
> [R4-A3: Additional baselines for MVTec data]:
> As we explained above, our paper focuses on a data-centric approach to unsupervised anomaly detection, and we do not claim that our method is the state-of-the-art across all semi-supervised anomaly detection settings (i.e., all the training samples are normal as in the link). We claim that any anomaly detection models developed for semi-supervised settings (including the current state-of-the-art anomaly detection models) can benefit from the proposed SRR framework by being more robust when extended to fully-unsupervised settings (i.e. when there are anomalous/contaminated samples in the training data). In addition, we show the generalizability of our SRR framework with various anomaly detection models (such as CutPaste, Contrastive, GOAD) in the current manuscript, to emphasize on the generalizability of our ideas.
>
> [R4-A4: Publishing the codebase]:
> Thanks for your positive comments about our work. We hope that our points clarify the novelty aspects of SRR. Specifically, we would like to recap that SRR is not a direct competitor to GOAD, and indeed its key idea of being data refinement based, is very different from the key idea of GOAD. Instead, SRR can be applied on top of GOAD to improve its performance in truly unsupervised settings (i.e. not assuming only normal data is used during training as the papers in the link you share). Thus, we believe our paper would fill an important gap in the anomaly detection literature and we hope these address all your concerns.
>
> Our codebase will be published upon acceptance, and we would ensure full reproducibility. Regarding the time cost, we have added further analysis. We would like to mention that all the experiments are done on a single V100 GPU and each experiment needs at most 12 hours for training (the additional training time caused by the SRR framework is an average 13.1% of the total training time). Note that the computations of the ensemble parts can be further improved by the model parallelization.

---

### Author Response · Authors · 2021-11-20
**Answers to all reviewers**

Thank you for reviewing our paper and providing insightful comments. We carefully read all the reviews and provided detailed answers in the replies below.

On the concern of novelty, we would like to highlight that our method distinguishes from the literature by bringing a data-centric approach (refining the unlabeled data) to unsupervised anomaly detection beyond the model-centric approaches (improving the model itself such as RobustPCA and Robust Autoencoder). Particularly, our paper is the only one that shows significant unsupervised anomaly detection performance improvements by iterative employment of data refinement (with a selection mechanism based on an ensemble learning) along with self-supervised learning for better representations to employ anomaly detection on. Most anomaly detection models are not designed for truly unsupervised settings, but rather semi-supervised settings, as most assume that all the training samples are normal (which indeed requires human supervision to construct). We demonstrate the importance of accurate data refinement to obtain better representations with self-supervised learning, and thus improved anomaly detection performance in fully unsupervised settings - the design principle of SRR is matched with the truly unsupervised anomaly detection objective. Note that any representation learning method (beyond CutPaste, Contrastive, and GOAD in the manuscript) and any OCC model (beyond OC-SVM etc.) can be combined with SRR framework for unsupervised anomaly detection. Please check more detailed answers in the replies below.

---

### Decision · Program_Chairs · 2022-01-20

**Decision:**

Reject

**Comment:**

The paper worked on fully unsupervised anomaly detection and proposed to use self-supervised representation learning to improve the performance of one-class classification. This is a borderline case close to acceptance but cannot make it. Specifically, it is useful, but its novelty is the main issue, since it is not surprising that self-supervised representation learning can improve one-class classification without representation learning (this part is still much of the taste of ICLR) and an ensemble of multiple models can improve upon a single model (which is just "bootstrap aggregating" or "bagging" used everyday in practice and known to machine learning and statistics societies a very long time ago). After seeing the rebuttal, the concerns were not really addressed well and the issues were only partially solved. Thus, the paper is not enough to guarantee an acceptance to ICLR unfortunately.